# Generalized Balancing Weights via Deep Neural Networks

## Abstract

Estimating causal effects from observational data is a central problem in many
domains. A general approach is to balance covariates with weights such that the
distribution of the data mimics randomization. We present generalized balanc-
ing weights, *Neural Balancing Weights* (NBW), to estimate the causal effects of
an arbitrary mixture of discrete and continuous interventions. The weights were
obtained through direct estimation of the density ratio between the source and bal-
anced distributions by optimizing the variational representation of $f$-divergence.
For this, we selected $\alpha$-divergence as it presents efficient optimization because
it has an estimator whose sample complexity is independent of its ground truth
value and unbiased mini-batch gradients; moreover, it is advantageous for the
vanishing-gradient problem. In addition, we provide the following two methods
for estimating the balancing weights: improving the generalization performance
of the balancing weights and checking the balance of the distribution changed by
the weights. Finally, we discuss the sample size requirements for the weights as
a general problem of a curse of dimensionality when balancing multidimensional
data. Our study provides a basic approach for estimating the balancing weights of
multidimensional data using variational $f$-divergences.

## 1 Introduction

Estimating causal effects from observational data is a central problem in many application domains,
including public health, social sciences, clinical pharmacology, and clinical decision-making. One
standard approach is balancing covariates with weights that are the same as the density ratios be-
tween the source and balanced distributions, such that their distribution mimics randomization.
Many methods have been developed to estimate the balancing weights, such as inverse propen-
sity weighting (IPW) Rosenbaum and Rubin [24], augmented inverse propensity weighting (AIPW)
[22], generalized propensity score (GPS) [10], covariate balancing propensity score (CBPS) [9],
overlap weighting [13], and entropy balancing (EB) [8, 30]. However, these methods are limited to
categorical or continuous interventions.

In this study, we propose generalized balancing weights to estimate the causal effects of an arbitrary
mixture of discrete and continuous interventions. To the best of our knowledge, no causal infer-
ence method focusing on the balancing weights exists for this problem. We approach this problem
by directly estimating the density ratio, more precisely, the Radon–Nikodým derivatives, between
the source and balanced distributions using a neural network algorithm by optimizing a variational
representation of a $f$-divergence. $f$-divergences, whose values are greater than or equal to zero and
considered zero if the two distributions are equal, are the statistics used to measure the closeness
of the two distributions. The optimal functions for the variational representations derived from $f$-
divergences with the Legendre transform correspond to the density ratio between the distributions

[16]. An approach to estimate the density ratio by optimizing a variational representation of a $f$-divergence was developed in the domain adaptation region [29].

However, optimizing the $f$-divergences, including estimating the density ratio, is challenging. This is due to the following reasons. First, for KL-divergence, the dominant $f$-divergence, the requirements for sample size increase exponentially with the true amount of the divergence [14, 28]. Second, a naive gradient estimate over mini-batch samples leads to a biased estimate of the full gradient [4]. Third, gradients of neural networks often vanish when the estimated probability ratios are close to zero [2].

To avoid the first problem, we focus on $\alpha$-divergence, which is a subgroup of $f$-divergence. $\alpha$-divergence has an estimator whose sample complexity is independent of its ground truth value and unbiased mini-batch gradients. In addition, by selecting $\alpha$ from a particular interval, we avoid vanishing gradients of neural networks when the neural networks reach extreme local minima.

In addition, we provide two techniques for estimating the balancing weights. First, we propose a validation method using test data and an early stopping method to improve the generalization performance of balancing. The generalization performance of the weights worsens as the dimensions of the data increase, and the sample size requirements of the weights increase exponentially with the dimensions. Next, we present a method for measuring the performance of balancing weights by estimating the $\alpha$-divergence information to check the balance of the distribution,

This study is divided into seven parts. First, we introduce the background of the study. Second, we review related studies. Third, we define the terminology and concepts for causal inferences. Fourth, we present our novel method for estimating balancing weights. Fifth, we provide techniques for estimating the weights. Sixth, we discuss the sample requirements for the weights. Finally, we conclude this paper. All the numerical experiments and proofs are described in the appendix.

## 2   Related Work

**Balancing weight: Balancing weight:** Many methods have been proposed to estimate the balancing weights. The following methods are proposed for binary intervention: IPW [24], AIPW [22], CBPS [9], and overlap weighting [13]. The following methods have been proposed for continuous intervention: GPS [10] and EB [8, 30]. **Statistical divergences and density ratio estimation:** Despite the abundance of classic studies [15, 29], we focused on studies that directly estimate density ratios or optimize statistical divergences using neural networks. In this review, these studies have beenclassified into four groups. First is the estimation of KL-divergence or mutual information [3, 18, 21]; the second is density ratio estimation [11]; the third is generative adversarial networks (GANs) [17, 31, 6, 32] (statistical divergences were used as discriminators for GANs); and the fourth is domain generation [27, 6, 35, 1]. In addition to these application studies, divergences were improved [5].

## 3   Terminologies and Definitions

Here, we briefly introduce the terminology and definitions used in this study.

**Notations and Terminologies.**   Random variables are denoted by capital letters; for example, $A$. Small letters are used for the values of random variables of the corresponding capital letters; $a$ is the value of the random variable $A$. Bold letters $\mathbf{A}$ or $\mathbf{a}$ represent a set of variables or random variable values. In particular, $\mathbf{V} = \{V_1, \ldots, V_n\}$ are used for the observed random variables and $\mathbf{U} = \{U_1, \ldots, U_m\}$ are used as unobserved random variables. For example, the domain of the variable $A$ is denoted by $\mathcal{X}_A$, and $\mathcal{X}_{A_1} \times \cdots \times \mathcal{X}_{A_n}$ is denoted by $\mathcal{X}_\mathbf{A}$ for $\mathbf{A} = A_1 \times \cdots \times A_n$. $\mathbf{V} \cup \mathbf{U}$ are assumed to be semi-Markovian models and $G = G_{\mathbf{VU}}$ denotes the causal graph for $\mathbf{V} \cup \mathbf{U}$. $Pa(\mathbf{A})_G$, $Ch(\mathbf{A})_G$, $An(\mathbf{A})_G$, and $De(\mathbf{A})_G$ represent parents, children, ancestors, and descendants of the observed variables in $G$, respectively, for $\mathbf{A} \subset \mathbf{V}$. In this study, $Pa(\mathbf{A})_G$, $Ch(\mathbf{A})_G$, $An(\mathbf{A})_G$, and $De(\mathbf{A})_G$ do not include $\mathbf{A}$. $P$ and $Q$ are used as the probability measures on $(\mathbb{R}^d, \mathscr{F})$, where $\mathscr{F}$ denotes the $\sigma$-algebra of subsets of $\mathbb{R}^d$. $E_P[\cdot]$ and $E_P[\cdot|\cdot]$ denote expectation and conditional expectation under the distribution $P$, respectively. For example, $E_P[\mathbf{X}] = \int_{\mathcal{X}_\mathbf{X}} dP$ and $E_P[\mathbf{Y}|\mathbf{X}] = \int_{\mathcal{X}_\mathbf{Y}} dP(\mathbf{Y}|\mathbf{X})$. $\hat{E}_P[\cdot]$ denotes the empirical expectation under $P$; that is, the

sample mean of the finite observations drawn from $P$. $P$ is called *absolute continuous* with respect to $Q$, $P(A) = 0$ whenever $Q(A) = 0$ for any $A \in \mathscr{F}$, which is represented as $P \ll Q$. $\frac{dP}{dQ}$ denotes the Radon–Nikodým derivative of $P$ with respect to $Q$ for $P$ and $Q$ with $P \ll Q$. In this study, we refer to density ratios as the Radon–Nikodým derivatives. $\mu$ denotes a probability measure on $\mathbb{R}^d$ with $P \ll \mu$ and $Q \ll \mu$. $\mathbf{X}^{(N)} = \{\mathbf{X}^1, \ldots, \mathbf{X}^N\}$ denotes $N$ i.i.d. random variables from $\mu$. $\mathbf{X}_P^{(N)} = \{\mathbf{X}_{\sim P}^1, \ldots, \mathbf{X}_{\sim P}^N\}$ and $\mathbf{X}_Q^{(N)} = \{\mathbf{X}_{\sim Q}^1, \ldots, \mathbf{X}_{\sim Q}^N\}$ denote variables defined as $P(\mathbf{X}_{\sim P}^i \leq \mathbf{x}) = \mu(\mathbf{X}^i \leq \mathbf{x})$ and $Q(\mathbf{X}_{\sim Q}^i \leq \mathbf{x}) = \mu(\mathbf{X}^i \leq \mathbf{x})$, $\forall \mathbf{x} \in \mathbb{R}^d$, for $1 \leq i \leq N$. We represent $f \lesssim g$ when $\lim\sup_{n\to\infty} f(n)/g(n) < \infty$ holds. The notation $f \gtrsim g$ is defined similarly.

## 3.1 Definitions

In this study, we considered the causal effects of joint and multidimensional interventions. For clarity, we used different notations, "$do$" and "$\overline{do}$," for single-dimensional and multidimensional interventions, respectively. [1] For a single-dimensional intervention, a $do$ symbol is used, which is the same as Pearl's $do$-calculation.

**Definition 3.1** ($do$-calculation, Pearl(2009)). For the two given disjoint sets of $\mathbf{X}, \mathbf{Y} \subset \mathbf{V}$, the causal effect on $\mathbf{Y}$ for intervention in $\mathbf{X}$ with values $\mathbf{x}$, denoted by $P(\mathbf{Y}|do(\mathbf{X} = \mathbf{x}))$, is defined as the probability distribution, such that

$$P(\mathbf{Y}|do(\mathbf{X} = \mathbf{x})) = \sum_{\substack{\mathbf{v}' \in \mathcal{X}_{\mathbf{V}'} \\ pa_{\mathbf{x}} \in \mathcal{X}_{Pa(\mathbf{X})_G}}} \frac{P(\mathbf{Y}, \mathbf{X} = \mathbf{x}, Pa(\mathbf{X})_G = pa_{\mathbf{x}}, \mathbf{V}' = \mathbf{v}')}{P(\mathbf{X} = \mathbf{x}|Pa(\mathbf{X})_G = pa_{\mathbf{x}})}, \quad (1)$$

where $\mathbf{V}' = \mathbf{V} \setminus (\mathbf{X} \cup Pa(\mathbf{X})_G \cup \mathbf{Y})$. The causal effect of $\mathbf{X}$ on $\mathbf{Y}$ under the conditions $\mathbf{Z}$ denoted by $P(\mathbf{Y} = \mathbf{y}|do(\mathbf{X} = \mathbf{x}), \mathbf{Z} = \mathbf{z})$ is defined as the probability distribution, such that

$$P(\mathbf{Y} = \mathbf{y}|do(\mathbf{X} = \mathbf{x}), \mathbf{Z}) = \frac{P(\mathbf{Y} = \mathbf{y}, \mathbf{Z}|do(\mathbf{X} = \mathbf{x}))}{P(\mathbf{Z}|do(\mathbf{X} = \mathbf{x}))}. \quad (2)$$

Notably, from Definition 3.1, a $do$-calculation for a set of variables coincides with the simultaneous interventions for each variable:

$$P(\mathbf{Y}|do(\mathbf{X})) = P(\mathbf{Y}|do(X_1), do(X_2), \ldots, do(X_n)), \quad (3)$$

where $\mathbf{X} = \{X_1, X_2, \ldots, X_n\}$. Here, we refer to each intervention in (3) as a "single-dimensional intervention".

Furthermore, we use the $\overline{do}$ symbol for multidimensional intervention. Intuitively, a $\overline{do}$ symbol represents the intervention of the variables that preserves the functional relationship within the variables.

**Definition 3.2** ($\overline{do}$ symbol). $\overline{do}$ symbol defines the following probability distribution:

$$P(\mathbf{Y}|\overline{do}(\mathbf{X_1}), \overline{do}(\mathbf{X_2}), \ldots, \overline{do}(\mathbf{X_n})) = P(\mathbf{Y}|do(\mathbf{X})) \times P(\mathbf{X_1}) \times P(\mathbf{X_2}) \times \cdots \times P(\mathbf{X_n}), \quad (4)$$

where $\mathbf{X} = \mathbf{X}_1 \cup \mathbf{X}_2 \cup \cdots \cup \mathbf{X}_n$.

$\overline{do}$ symbols are useful, particularly when we consider interventions in a multivalued discrete variable expressed using one-hot encoding. In this case, we cannot express the causal effect effectively using $do$ symbols. For example, let us consider the case of an intervention in the ternary variable $X$, $\mathcal{X}_X = \{x_1, x_2, x_3\}$ and let $X$ be expressed by $\mathbf{X}' = (X_1', X_2', X_3')$, such that $X_i' = 1$ if $X = \mathbf{x}_i$ otherwise $X_i' = 0$ for $i = 1, 2, 3$. Then, $P(\cdot|do(X = \mathbf{x}_3))$ is the same as $P(\cdot|\overline{do}(\mathbf{X}' = (0, 0, 1)))$, which differs from $P(\cdot|do(\mathbf{X}' = (0, 0, 1)))$. We refer to this type of intervention as a "multidimensional intervention".

Next, we provide definitions of the $f$-divergence and $f$-divergence information.

**Definition 3.3** ($f$-divergence). The $f$-divergence $D_f$ between the two probability measures $P$ and $Q$ with $Q \ll P$ induced by a convex function $f$ satisfying $f(1) = 0$ is defined by $D_f(Q||P) = E_P[f(dQ/dP)]$.

---

[1]The values of the variables in the parentheses for both symbols can be dropped if not necessary in the context. For example, we sometimes represent $do(\mathbf{X} = \mathbf{x})$ or $\overline{do}(\mathbf{X} = \mathbf{x})$ as $do(\mathbf{X})$ or $\overline{do}(\mathbf{X})$, respectively.

Many divergences are specific cases obtained by selecting a suitable generator function $f$. For example, $f(u) = u \log u$ corresponds to the KL-divergence. In particular, we focus on $\alpha$-divergence, which is expressed as follows:

$$D_\alpha(Q||P) = E_P \left[ \frac{1}{\alpha(\alpha-1)} \left\{ \left( \frac{dQ}{dP} \right)^{1-\alpha} - 1 \right\} \right], \tag{5}$$

where $\alpha \in \mathbb{R} \setminus \{0, 1\}$. From (5), Hellinger divergence is obtained as $\alpha = 1/2$, and $\chi^2$ divergence by $\alpha = -1$.

From $f$-divergence, the $f$-divergence information is defined as the mutual information if we choose the KL-divergence as the $f$-divergence. Here, we present a definition of $f$-divergence information for multi-variables.

**Definition 3.4** ($f$-divergence information)**.** For disjoint variables $\mathbf{X} = \{\mathbf{X}_1, \mathbf{X}_2, \ldots, \mathbf{X}_n\} \subset \mathbf{V}$, let $P_\mathbf{X}$ be the joint probability measure for $\mathbf{X}$. For each $i = 1, 2, \ldots, n$, $P_{\mathbf{X}_i} = \int_{\mathcal{X}_{\mathbf{X} \setminus \mathbf{X}_i}} dP_\mathbf{X}$ is a measure of the marginal distribution of $P_\mathbf{X}$ for $\mathbf{X}_i$. The $f$-divergence information for $\mathbf{X}_1, \mathbf{X}_2, \ldots, \mathbf{X}_n$ under $P_\mathbf{X}$ and a convex function $f$ satisfying $f(1) = 0$ is defined as the $f$-divergence between $P_\mathbf{X}$ and $P_{\mathbf{X}_1} \times P_{\mathbf{X}_2} \times \cdots \times P_{\mathbf{X}_n}$:

$$I_f(\mathbf{X}_1, \mathbf{X}_2, \ldots, \mathbf{X}_n; P_\mathbf{X}) = E_{P_\mathbf{X}} \left[ f \left( \frac{dP_{\mathbf{X}_1} \times dP_{\mathbf{X}_2} \times \cdots \times dP_{\mathbf{X}_n}}{dP_\mathbf{X}} \right) \right]. \tag{6}$$

# 4 Problem Set Up

Before describing the details of the problem, we provide a notation for the probability distribution, which is the goal of balancing. Hereafter, $P$ denotes the probability distribution of observational data. For the given disjoint sets $\mathbf{X}_1, \mathbf{X}_2, \ldots, \mathbf{X}_n, \mathbf{Y}, \mathbf{Z} \subset \mathbf{V}$, let $\widetilde{P}$ be a probability distribution, as follows:

$$\begin{aligned} \widetilde{P} &= P(\mathbf{Y}|\overline{do}(\mathbf{X_1}), \overline{do}(\mathbf{X_2}), \ldots, \overline{do}(\mathbf{X_n}), \mathbf{Z}) \times P(\mathbf{Z}) \\ &= P(\mathbf{Y}|do(\mathbf{X}), \mathbf{Z}) \times P(\mathbf{X_1}) \times P(\mathbf{X_2}) \times \cdots \times P(\mathbf{X_n}) \times P(\mathbf{Z}), \end{aligned} \tag{7}$$

where $\mathbf{X} = \mathbf{X}_1 \cup \mathbf{X}_2 \cup \cdots \cup \mathbf{X}_n$. $\widetilde{P}$ is the probability distribution of the counterfactual data from simultaneous (multidimensional) interventions in $\mathbf{X_1}, \mathbf{X_2}, \ldots, \mathbf{X_n}$ under the condition $\mathbf{Z}$.

**Objective.** The objective of this study is to obtain the balancing weights that transform $P(\mathbf{Y}, \mathbf{X}, \mathbf{Z})$ into $\widetilde{P}(\mathbf{Y}, \mathbf{X}, \mathbf{Z})$. More precisely, given the i.i.d. observational data $\{(\mathbf{x}^i, \mathbf{z}^i)|i = 1, 2, \ldots, N\}$, we aim to estimate the weights $BW(\mathbf{X}, \mathbf{Z})$, such that

$$E_{\widetilde{P}}[f(\mathbf{X}, \mathbf{Z})] = E_P[f(\mathbf{X}, \mathbf{Z}) \cdot BW(\mathbf{X}, \mathbf{Z})] \tag{8}$$

holds for any measurable function $f$ on $\mathbb{R}^d$. If we obtain the weights, we estimate the conditional average causal effect (CACE) for $P(\mathbf{Y}|\overline{do}(\mathbf{X_1}), \overline{do}(\mathbf{X_2}), \ldots, \overline{do}(\mathbf{X_n}), \mathbf{Z})$, that is $E_{\widetilde{P}}[\mathbf{Y}|\mathbf{X}, \mathbf{Z}]$, using state-of-the-art supervised machine learning algorithms, with the weights assigned as the individual weights for each sample.

**Assumptions.** We assumed the following to achieve our objective:

- Assumption 1. The causal effect $P(\mathbf{Y}|do(\mathbf{X}))$ is identifiable, or equivalently, $\widetilde{P}$ from (7) can be identified. [2][3]

- Assumption 2. Let $\mathbb{P} = P(\mathbf{X}_1, \mathbf{X}_2, \ldots, \mathbf{X}_n, \mathbf{Z})$ and let $\mathbb{Q} = P(\mathbf{X_1}) \times P(\mathbf{X_2}) \times \cdots \times P(\mathbf{X_n}) \times P(\mathbf{Z})$. Subsequently, we assume that $\mathbb{Q} \ll \mathbb{P}$.

Assumption 2 is the same as the *overlap* assumption if we consider this a single-dimensional intervention. Here, we propose overlapped assumptions for joint and multidimensional interventions.

---

[2]The simplest case that satisfies Assumption 1 is that no confounding exists among the data ([20], P78, Theorem 3.2.5).

[3]If certain unobserved data are assumed to exist, the identifiability of the causal effect is determined by the structure of the causal diagram for $P$. One criterion for the identifiability of a causal effect is expressed by [26]. The discussion of the identifiability of the causal effect is beyond the scope of this study.

## 5 Estimation of Balancing Weights

In this section, we present the way to effectively estimate the probability density ratios by optimizing $f$-divergence.

**Density Ratios as Balancing Weights.** We first note that the density ratios, which are referred to as the Radon–Nikodým derivative in this paper, are equal to the balancing weight of the target. For a density ratio of $P$ to $\widetilde{P}$, that is $\frac{d\widetilde{P}}{dP}$, it holds that

$$E_{\widetilde{P}}[f] = \int f \cdot \frac{d\widetilde{P}}{dP} \cdot dP = E_P\left[f \cdot \frac{d\widetilde{P}}{dP}\right], \tag{9}$$

for any measurable function $f$ in $\mathbb{R}^d$. Then, (8) and (9) are equivalent. As an example of the aforementioned density ratio, let $X$ be a binary variable with $\mathcal{X}_X = \{1,0\}$ and let $\mathbf{Z}$ be covariates. Using propensity score $e(\mathbf{z}) = P(X = 1|\mathbf{Z} = \mathbf{z})$, we observe that $\frac{d\widetilde{P}}{dP}(X = 1, \mathbf{z}) = P(X = 1)/e(\mathbf{z})$ and $\frac{d\widetilde{P}}{dP}(X = 0, \mathbf{z}) = P(X = 0)/(1-e(\mathbf{z}))$. That is, $\frac{d\widetilde{P}}{dP}$ is the stabilized inverse probability of the treatment weighting [23].

### 5.1 Our Approach

Our approach involves obtaining the density ratios as an optimal function for a variational representation of an $f$-divergence. This approach is based on the fact that the optimal function is connected to density ratios [15].

**Variational representation.** Using the Legendre transform of the convex conjugate of a twice differentiable convex function $f$, $f^*(\psi) = \sup_{r\in\mathbb{R}}\{\psi\cdot r - f(r)\}$, we obtain a variational representation of $f$-divergence:

$$D_f(Q||P) = \sup_{\phi\geq 0}\{E_Q[f'(\phi)] - E_P[f^*(f'(\phi))]\}, \tag{10}$$

where supremum is considered over all measurable functions with $E_Q[f'(\phi)] < \infty$ and $E_P[f^*(f'(\phi))] < \infty$. The maximum value is achieved at $\phi = dQ/dP$.

We obtained the optimal function for (10) by replacing $\phi$ in the equation with a neural network model $\phi_\theta$ and training it through back-propagation with a loss function, such that

$$\mathcal{L}(\theta) = -\left\{\hat{E}_Q[f'(\phi_\theta)] - \hat{E}_P[f^*(f'(\phi_\theta))]\right\}. \tag{11}$$

**Selecting $\alpha$-divergence for Optimization.** We select $\alpha$-divergence for the following reasons. First, the sample size requirements for $\alpha$-divergence is independent of its ground truth value: second, it has unbiased mini-batch gradients; third, it can avoid a vanishing gradient problem.

The variational representation of $\alpha$-divergence is as follows (Lemma C.1 in Appendix C):

$$D_\alpha(Q||P) = \sup_{\phi\geq 0}\left\{\frac{1}{\alpha(1-\alpha)} - \frac{1}{\alpha}E_Q\left[\phi^{-\alpha}\right] - \frac{1}{1-\alpha}E_P\left[\phi^{1-\alpha}\right]\right\}. \tag{12}$$

**Sample size requirements for $\alpha$-divergence.** The $\alpha$-divergence has an estimator with sample complexity $O(1)$ (Corollary 1 in Birrell et al., 2022, P19; Corollary C.10 in Appendix C). Conversely, the sample complexity of KL-divergence is $O(e^{KL(Q||P)})$ [14, 28]:

$$\lim_{N\to\infty} \frac{N \cdot \text{Var}\left[\widehat{KL^N}(Q||P)\right]}{KL(Q||P)^2} \geq \frac{e^{KL(Q||P)} - 1}{KL(Q||P)^2}, \tag{13}$$

where $\widehat{KL^N}(Q||P)$ is the KL-divergence estimator for sample size N using a variational representation of the divergence, and $KL(Q||P)$ is the ground truth value.

**Unbiasedness for mini-batch gradients.** $\phi$ in (12) can be expressed in a Gibbs density form (Proposition C.2 in Appendix C). Then, we observe that

$$D_\alpha(Q||P) = \sup_T \left\{ \frac{1}{\alpha(1-\alpha)} - \frac{1}{\alpha} E_Q\left[e^{\alpha \cdot T}\right] - \frac{1}{1-\alpha} E_P\left[e^{(\alpha-1)\cdot T}\right] \right\}, \qquad (14)$$

where supremum is considered over all measurable functions $T : \mathbb{R}^d \to \mathbb{R}$ with $E_P[e^{(\alpha-1)\cdot T}] < \infty$ and $E_Q[e^{\alpha \cdot T}] < \infty$.

From this equation, we obtain our loss function, which has unbiasedness for mini-batch gradients (Proposition C.8 in Appendix C), as follows :

$$\mathcal{L}_\alpha(\theta) = \frac{1}{\alpha} \hat{E}_Q\left[e^{\alpha \cdot T_\theta}\right] + \frac{1}{1-\alpha} \hat{E}_P\left[e^{(\alpha-1)\cdot T_\theta}\right]. \qquad (15)$$

**Advantage in vanishing gradients problem.** By setting $\alpha$ within $(0, 1)$, we can avoid vanishing gradients of neural networks when they reach the extreme local minima. The vanishing-gradient problem for optimizing divergence is known in GANs [2]. Now, we consider the case where the probability ratio $e^{T_\theta(\mathbf{x})}$ in (15) is nearly zero or large for some point $\mathbf{x}$, corresponding to cases in which the probabilities for $P$ or $Q$ at some points are much smaller than those for the other.

To show the relation between $e^{T_\theta(\mathbf{x})}$ and the learning of the neural networks, we obtain gradient of (15):

$$\nabla_\theta \mathcal{L}_\alpha(\theta) = \hat{E}_Q\left[\nabla_\theta T_\theta \cdot e^{\alpha \cdot T_\theta}\right] - \hat{E}_P\left[\nabla_\theta T_\theta \cdot e^{(\alpha-1)\cdot T_\theta}\right]. \qquad (16)$$

The behavior of $\nabla_\theta \mathcal{L}_\alpha(\theta)$ when $E_Q[e^{T_\theta}] \to 0$ or $E_Q[e^{T_\theta}] \to \infty$, under some regular conditions for $T_\theta$ and an assumption that $P \ll Q$, can be summarized as follows: Let $E[\,\cdot\,]$ denote $E_P[E_Q[\,\cdot\,]]$, then

$\alpha > 1$: $\quad E[\nabla_\theta \mathcal{L}_\alpha(\theta)] \to \vec{0}$ (as $E_Q[e^{T_\theta}] \to 0$), and $\quad E[\nabla_\theta \mathcal{L}_\alpha(\theta)] \to \vec{\infty} - \vec{\infty}$ (as $E_Q[e^{T_\theta}] \to \infty$).

$\alpha < 0$: $\quad E[\nabla_\theta \mathcal{L}_\alpha(\theta)] \to \vec{0}$ (as $E_Q[e^{T_\theta}] \to \infty$), and $\quad E[\nabla_\theta \mathcal{L}_\alpha(\theta)] \to \vec{\infty} - \vec{\infty}$ (as $E_Q[e^{T_\theta}] \to 0$).

$0 < \alpha < 1$: $E[\nabla_\theta \mathcal{L}_\alpha(\theta)] \to -\vec{\infty}$ (as $E_Q[e^{T_\theta}] \to 0$), and $\quad E[\nabla_\theta \mathcal{L}_\alpha(\theta)] \to \vec{\infty}$ (as $E_Q[e^{T_\theta}] \to \infty$).

Notably, $E_Q[e^{T_\theta}] \to 0 \Leftrightarrow E_P[e^{T_\theta}] \to 0$ $E_Q[e^{T_\theta}] \to \infty \Leftrightarrow E_P[e^{T_\theta}] \to \infty$, because $Q \ll P$ and $P \ll Q$.

For $\alpha > 1$ and $\alpha < 0$, cases exist where $E[\nabla_\theta \mathcal{L}_\alpha(\theta)] \to \vec{0}$. This implies the possibility that the neural networks reach extreme local minima such that their estimations for density ratios are 0 or $\infty$. However, this problem can be avoided by selecting $\alpha$ from interval $(0, 1)$. We note that the selecting of $\alpha$ does not cause instability in numerical calculations for cases where $E[\nabla_\theta \mathcal{L}_\alpha(\theta)] \to \vec{\infty} - \vec{\infty}$. In Appendix D.1, we present numerical experimental results for different values of $\alpha$.

# 6 Method

In this section, we first present the main theorem that summarizes the new balancing weight method proposed herein. Next, we present the balancing weight method.

## 6.1 Main Theorem

Here, we present the main theorem that summarizes the new balancing weight method proposed herein.

**Theorem 6.1.** *Given disjoint sets of* $\mathbf{X} = \{\mathbf{X}_1, \mathbf{X}_2, \ldots, \mathbf{X}_n\}, \mathbf{Y}, \mathbf{Z} \subset \mathbf{V}$ *satisfying*

$$\mathbf{X} = \{\mathbf{X}_1, \mathbf{X}_2, \ldots, \mathbf{X}_n\} \subset An(\mathbf{Y})_G \quad and \quad \mathbf{Z} \cap De(\mathbf{Y})_G = \phi. \qquad (17)$$

*Let* $\mathbb{P} = P(\mathbf{X}_1, \mathbf{X}_2, \ldots, \mathbf{X}_n, \mathbf{Z})$ *and* $\mathbb{Q} = P(\mathbf{X_1}) \times P(\mathbf{X_2}) \times \cdots \times P(\mathbf{X_n}) \times P(\mathbf{Z})$, *and* $\widetilde{P} = P(\mathbf{Y}|do(\mathbf{X}), \mathbf{Z}) \times P(\mathbf{X_1}) \times P(\mathbf{X_2}) \times \cdots \times P(\mathbf{X_n}) \times P(\mathbf{Z})$. *We assume that* $P$ *satisfies Assumptions 1 and 2 in the aforementioned setting, and it holds that* $E_\mathbb{P}\left[(d\mathbb{Q}/d\mathbb{P})^{1-\alpha}\right] < \infty$ *for some* $0 < \alpha < 1$, *then, for the optimal function* $T^*$, *such that*

$$T^*(\mathbf{X}_1, \mathbf{X}_2, \ldots, \mathbf{X}_n, \mathbf{Z}) = \arg\inf_{T \in \mathcal{T}^\alpha} \left\{ \frac{1}{\alpha} E_\mathbb{Q}\left[e^{\alpha \cdot T}\right] + \frac{1}{1-\alpha} E_\mathbb{P}\left[e^{(\alpha-1)\cdot T}\right] \right\}, \qquad (18)$$

**Algorithm 1** Training a Neural Balancing Weight model

---

**Input:** Train Data $(\mathbf{x}_1, \{(\mathbf{x}_1^i, \ldots, \mathbf{x}_n^i, \mathbf{z}^i)\}_{i=1}^N$
**Output:** A Neural Balancing Weight Model
$T_{\theta_K}$
$\sigma_1^{\mathbf{x}} \leftarrow \mathtt{SHUFFLE}(\{1:N\})$
$\quad\quad\quad\vdots$
$\sigma_n^{\mathbf{x}} \leftarrow \mathtt{SHUFFLE}(\{1:N\})$
$\sigma^{\mathbf{z}} \leftarrow \mathtt{SHUFFLE}(\{1:N\})$

**for** $t = 1$ to $K$ **do**
$\quad \hat{E}_{\mathbb{P}} \leftarrow \frac{1}{N} \Sigma_{i=1}^N e^{(\alpha-1) \cdot T_{\theta_t}(\mathbf{x}_1^i, \ldots, \mathbf{x}_n^i, \mathbf{z}^i)}$
$\quad \hat{E}_{\mathbb{Q}} \leftarrow \frac{1}{N} \Sigma_{i=1}^N e^{\alpha \cdot T_{\theta_t}(\mathbf{x}_1^{\sigma_1^{\mathbf{x}}(i)}, \ldots, \mathbf{x}_n^{\sigma_n^{\mathbf{x}}(i)}, \mathbf{z}^{\sigma^{\mathbf{z}}(i)})}$

$\quad \mathcal{L}_\alpha(\theta_t) \leftarrow \hat{E}_{\mathbb{Q}}/\alpha + \hat{E}_{\mathbb{P}}/(1-\alpha)$
$\quad \theta_{t+1} \leftarrow \theta_t - \nabla_{\theta_t} \mathcal{L}_\alpha(\theta_t)$
**end for**

---

*it holds that*

$$\frac{d\widetilde{P}}{dP} = e^{-T^*(\mathbf{X}_1, \mathbf{X}_2, \ldots, \mathbf{X}_n, \mathbf{Z})}. \tag{19}$$

*Here, $\mathcal{T}^\alpha$ denotes the set of all non-constant functions $T(\mathbf{x}) : \mathbb{R}^d \to \mathbb{R}$ with $E_{\mathbb{P}}[e^{(\alpha-1) \cdot T(\mathbf{X})}] < \infty$.*

*Proof.* See Appendix C. □

Here, we mention that the assumption (17) is necessary for the (19) to hold, which is derived from our Theorem C.15 in Appendix C.

## 6.2 Balancing Weight Method

We present the implementation of training a neural balancing weights (NBW) model in Algorithm 1. It is important to consider the stopping time $K$ for neural network model $T_{\theta_K}$ in Algorithm 1, which is discussed in the next section. To obtain the sample mean under $\mathbb{Q}$, that is, the estimator for $E_{\mathbb{Q}}[e^{\alpha \cdot T_\theta}]$ in (18), a shuffling operation can be used for the samples. Now, we define neural balancing weights (NBW). [4] [5]

**Definition 6.2** (Neural Balancing Weights). Let $T_{\theta_K}$ be a neural networks obtained from Algorithm 1. Then, the NBW of $T_{\theta_K}$, expressed as $BW(\mathbf{X}_1, \mathbf{X}_2, \ldots, \mathbf{X}_n, \mathbf{Z}; T_{\theta_K})$, are defined as

$$BW(\mathbf{X}_1, \mathbf{X}_2, \ldots, \mathbf{X}_n, \mathbf{Z}; T_{\theta_K}) = \frac{1}{Z} e^{-T_{\theta_K}(\mathbf{X}_1, \mathbf{X}_2, \ldots, \mathbf{X}_n, \mathbf{Z})}, \tag{20}$$

where $Z = \hat{E}_{\mathbb{P}}\left[e^{-T_{\theta_K}(\mathbf{X}_1, \mathbf{X}_2, \ldots, \mathbf{X}_n, \mathbf{Z})}\right]$.

We estimate $E_{\widetilde{P}}[\mathbf{Y}|\mathbf{X}, \mathbf{Z}]$, that is the CACE for $P(\mathbf{Y}|\overline{do}(\mathbf{X_1}), \overline{do}(\mathbf{X_2}), \ldots, \overline{do}(\mathbf{X_n}), \mathbf{Z})$, using $BW(\mathbf{X}_1, \mathbf{X}_2, \ldots, \mathbf{X}_n, \mathbf{Z}; T_{\theta_K})$ as the sample weights of the supervised algorithm:

$$\widehat{E}_{\widetilde{P}}[\mathbf{Y}|\mathbf{X}, \mathbf{Z}] = \widehat{E}_P[\mathbf{Y} \cdot BW_{\theta_K} \mid \mathbf{X}, \mathbf{Z}]. \tag{21}$$

Here, $\widehat{E}_P$ corresponds to the model of a supervised machine learning algorithm. As an example, we demonstrate a back-propagation algorithm using balancing weights for the mean squared error (MSE) loss in Algorithm 3 in Appendix E.

## 7 Techniques for NBW

We propose two techniques for estimating balancing weights: (i) improves generalization performance of the balancing weights. (ii) measures the performance of the balancing weights by estimating the $\alpha$-divergence information.

---

[4] We distinguish the notation of $BW(\cdot)$ by the expression of the variables in the parentheses. For example, for disjoint variables $X_1, X_2, X_3 \subset \mathbf{V}$, let $\mathbf{X} = \{X_1, X_2\}$. Then, $BW(\mathbf{X}, X_3; T_\theta)$ is used to indicate the balancing weights for $dP(X_1, X_2) \times dP(X_3)/dP(X_1, X_2, X_3)$. Conversely, $BW(X_1, X_2, X_3; T_\theta)$ denotes the balancing weights for $dP(X_1) \times dP(X_2) \times dP(X_3)/dP(X_1, X_2, X_3)$.

[5] However, we drop the variables in the parentheses and write $BW(\mathbf{X}_1, \mathbf{X}_2, \ldots, \mathbf{X}_n, \mathbf{Z}; T_\theta)$ as $BW_\theta$ if not necessary in the context.

## 7.1 Improving the Generalization Performance of the Balancing Weights

In this section, we first present an overfitting problem for balancing distributions. We then present two methods for improving the generalization performance of the weights: a validation method using test data and an early stopping method. Herein, let $T_{\theta_t}$ denote an NBW model at step $t$ in Algorithm 1. Let $\hat{\mathbf{X}}_Q^{(N)}(t) = e^{-T_{\theta_t}} \cdot \mathbf{X}_P^{(N)}$, that is, the data balanced by the weights of $e^{-T_{\theta_t}}$. Subsequently, let $\hat{Q}_t^{(N)}$ and $\hat{Q}^{(N)}$ denote the probability distributions of $\hat{\mathbf{X}}_Q^{(N)}(t)$ and $\hat{\mathbf{X}}_Q^{(N)}$, respectively, which correspond to the estimated and true distributions for balancing.

**An overfitting problem for balancing distributions.** From Corollary C.12 in Appendix C, we observe $\hat{\mathbf{X}}_Q^{(N)}(t) \xrightarrow{\mathrm{d}} \mathbf{X}_Q^{(N)}$ as $t \to \infty$. Then, Theorem 1 in [33] shows that

$$\lim_{t \to \infty} W_1(Q, \hat{Q}_t^{(N)}) = W_1(Q, \hat{Q}^{(N)}) \gtrsim N^{-1/(d-\delta)} \quad (\forall \delta > 0), \tag{22}$$

where $W_1$ is the Wasserstein distance of order 1 and $d$ is the lower Wasserstein dimension defined in [33]. (22) implies that, for balancing finite data, the destination of the balanced distribution is an empirical distribution, and the generalization performance of balancing worsens exponentially when the dimension of the data is larger. In view of optimizations of GANs, [34] referred to this phenomenon the " momorization " and proposed an early stopping method.

**Validation method using test data.** We can use a validation method using test data. Because $\hat{Q}^{(N)}$ and $\hat{P}^{(N)}$ are empirical probability distributions, we observe that $d\hat{Q}^{(N)}/d\hat{P}^{(N)}(x) = dQ/dP(x)$ if $x \in \mathbf{X}^{(N)}$, otherwise $d\hat{Q}^{(N)}/d\hat{P}^{(N)}(x) = 0$ (Proposition C.17 in Appendix C). Then, the optimal function of (15) for both distributions, that is $T_*^{(N)} = -log(d\hat{Q}^{(N)}/d\hat{P}^{(N)})$, is infinite except for the observations, and the loss of the $T_*^{(N)}$ is infinite for data independent of the observations. This implies that the loss of $T_t^{(N)}$ for the test data turns to increase from the middle of the training period, and we can determine the training step at which the generalization performance of the weights begins to worsen. In Section D.2 in Appendix D, we provide numerical experimental results to confirm the relationship between dimensions of data ($d$) and steps in training ($K$).

**Early stopping method.** In addition, we present an early stopping method for estimating the balancing weights as follows, which is inspired by the method developed in [34] (Corollary C.24 in Appendix C): for some $\delta > 0$, let

$$K_0 = C \cdot N^{2/(d+\delta)}, \tag{23}$$

where $C > 0$ is constant. Then, we have $W_1(Q, \hat{Q}_{K_0}^{(N)}) \lesssim N^{-1/(d+\delta)}$. Unfortunately, the curse of dimensionality remains in the proposed method. This will be discussed in the next section.

## 7.2 Measuring the Performance of the Balancing Weights

Let us assume that we obtain an NBW model $T_{\theta_0}$ and let $BW_{\theta_0} = BW(\mathbf{X}_1, \mathbf{X}_2, \ldots, \mathbf{X}_n, \mathbf{Z}; T_{\theta_0})$ be the balancing weights of $T_{\theta_0}$. If $BW_{\theta_0}$ successfully estimates $\frac{dQ}{dP}$, then the $\alpha$-divergence between $Q$ and $P_0$ will be nearly zero. Conversely, if $BW_{\theta_0}$ fails to estimate $\frac{dQ}{dP}$, the $\alpha$-divergence between $Q$ and $P_0$ is significantly different from zero. This implies that we can measure the performance of the balancing weights using the $\alpha$-divergence information for $P_0$.

Next, we present the definition of an $\alpha$-divergence information estimator using neural networks.

**Definition 7.1** (Neural $\alpha$-divergence Information Estimator). For disjoint variables $\mathbf{X}_1, \mathbf{X}_2, \ldots, \mathbf{X}_n \subset \mathbf{V}$, the neural $\alpha$-divergence information estimator for $P$ is defined as

$$\widehat{I}_\alpha(\mathbf{X}_1, \mathbf{X}_2, \ldots, \mathbf{X}_n; T_{\theta_*}) = \frac{1}{\alpha(1-\alpha)} - \inf_{\theta \in \Theta} \left\{ \frac{1}{\alpha} \hat{E}_Q \left[ e^{\alpha \cdot T_\theta} \right] + \frac{1}{1-\alpha} \hat{E}_P \left[ e^{(\alpha-1) \cdot T_\theta} \right] \right\}. \tag{24}$$

To measure the performance of balancing the weights from the NBW model, we estimate the $\alpha$-divergence information for balanced distribution from the weights. That is, we use the sample mean under a balanced distribution, despite the sample mean under $P$ for (24). For example, we assume

---

**Algorithm 2** Algorithm for checking the balance

---

**Input:** Train Data $\{(\mathbf{x}_1^i, \ldots, \mathbf{x}_n^i, \mathbf{z}^i)\}_{i=1}^N$, Test Data $\{(\widetilde{\mathbf{x}}_1^i, \ldots, \widetilde{\mathbf{x}}_n^i, \widetilde{\mathbf{z}}^i)\}_{i=1}^N$, A Neural Balancing Weight Model $T_\theta$

**Output:** The estimated $\alpha$-divergence information $\widehat{I}_\alpha$ for the balanced distribution with the balancing weights from $T_\theta$

$\sigma_1^{\mathbf{x}} \leftarrow \texttt{SHUFFLE}(\{1:N\})$

$\qquad \vdots$

$\sigma_n^{\mathbf{x}} \leftarrow \texttt{SHUFFLE}(\{1:N\})$
$\sigma^{\mathbf{z}} \leftarrow \texttt{SHUFFLE}(\{1:N\})$
$\{bw^i\}_i^N \leftarrow \dfrac{e^{-T_\theta(\mathbf{x}_1^i, \mathbf{x}_2^i, \ldots, \mathbf{x}_n^i, \mathbf{z}^i)}}{\sum\{e^{-T_\theta(\mathbf{x}_1^i, \mathbf{x}_2^i, \ldots, \mathbf{x}_n^i, \mathbf{z}^i)}\}}$
$\widehat{\mathbf{I}}_\alpha \leftarrow \{\}$

**for** $t = 1$ to $K$ **do**
$\quad \hat{E}_{\mathbb{P}_0} \leftarrow \frac{1}{N}\Sigma_{i=1}^N e^{(\alpha-1)\cdot T_\psi(\mathbf{x}_1^i, \ldots, \mathbf{x}_n^i, \mathbf{z}^i)} \cdot bw^i$
$\quad \hat{E}_{\mathbb{Q}} \leftarrow \frac{1}{N}\Sigma_{i=1}^N e^{\alpha\cdot T_\psi(\mathbf{x}_1^{\sigma_1^{\mathbf{x}}(i)}, \ldots, \mathbf{x}_n^{\sigma_n^{\mathbf{x}}(i)}, \mathbf{z}^{\sigma^{\mathbf{z}}(i)})}$

$\quad \mathcal{L}_\alpha(\psi) \leftarrow \hat{E}_{\mathbb{Q}}/\alpha + \hat{E}_{\mathbb{P}_0}/(1-\alpha)$
$\quad \psi \leftarrow \psi - \nabla_\psi \mathcal{L}_\alpha(\psi)$
$\quad \hat{E}_{\mathbb{P}_0}^{te} \leftarrow \frac{1}{N}\Sigma_{i=1}^N e^{(\alpha-1)\cdot T_\psi(\widetilde{\mathbf{x}}_1^i, \ldots, \widetilde{\mathbf{x}}_n^i, \widetilde{\mathbf{z}}^i)} \cdot bw^i$
$\quad \hat{E}_{\mathbb{Q}}^{te} \leftarrow \frac{1}{N}\Sigma_{i=1}^N e^{\alpha\cdot T_\psi(\widetilde{\mathbf{x}}_1^{\sigma_1^{\mathbf{x}}(i)}, \ldots, \widetilde{\mathbf{x}}_n^{\sigma_n^{\mathbf{x}}(i)}, \widetilde{\mathbf{z}}^{\sigma^{\mathbf{z}}(i)})}$
$\quad \widehat{I}_\alpha^t \leftarrow 1/\{\alpha \cdot (1-\alpha)\}$
$\qquad\qquad - \hat{E}_{\mathbb{Q}}^{te}/\alpha - \hat{E}_{\mathbb{P}_0}^{te}/(1-\alpha)$
$\quad \widehat{\mathbf{I}}_\alpha \leftarrow \widehat{\mathbf{I}}_\alpha \cup \{\widehat{I}_\alpha^t\}$
**end for**
$\widehat{I}_\alpha \leftarrow \max_t \widehat{\mathbf{I}}_\alpha$

---

that we have certain weights $BW' = \{bw^i : i = 1, 2, ..., N\}$, where $bw^i$ denotes the weight of sample $i$ of $N$. The balanced distribution from the weights is

$$dP' = BW' \cdot dP. \tag{25}$$

The $\alpha$-divergence information for $P'$ is estimated by replacing $P$ with $P'$ for (24) in the following manner: despite the sample mean $\hat{E}_P[e^{(\alpha-1)\cdot T_\theta}]$ for these equations, we use the weighted sample mean, such that

$$\hat{E}_{P'}[e^{(\alpha-1)\cdot T_\theta}] = \frac{1}{N}\Sigma_{i=1}^N bw^i \cdot e^{(\alpha-1)\cdot T_\theta(\mathbf{x}_1^i, \mathbf{x}_2^i, \ldots, \mathbf{x}_n^i, \mathbf{z}^i)}. \tag{26}$$

Details on the implementation for measuring the performance of balancing weights from an NBW model are provided in Algorithm 2, which includes the validation method for the overfitting problem in Section 7.1.

# 8 Limitations: Sample Size Requirements.

In Section 7.1, we noted that our method has a curse of dimensionality. The sample size requirement of the proposed method is $N > \left(\frac{1}{\varepsilon}\right)^{d+\delta}$ for $W_1(Q, \hat{Q}_{K_0}^{(N)}) < \varepsilon$ (Corollary C.25 in Appendix C). However, the curse of dimensionality is an essential problem when balancing multivariate data owing to the following factors. Because the optimal balancing weights defined as (8) for (finite) observational data are the density ratios of the empirical distributions, the distribution of the data balanced by them is the empirical distribution. Subsequently, owing to the balancing of the weights, the curse of dimensionality of the empirical distribution occurs, which is the same as that described in Section 7.1. Therefore, to achieve high generalization performance, we need to obtain weights that differ from the ideal density ratio between the source and target of the empirical distribution. Further research is required to address this problem. In Appendix D.3, we present the numerical examination results in which the causal effects of joint and multidimensional interventions were estimated with different sample sizes.

# 9 Conclusion

We propose generalized balancing weights to estimate the causal effects of an arbitrary mixture of discrete and continuous interventions. Three methods for training the weights were provided: an optimization method to learn the weights, a method to improve the generalization performance of the balancing weights, and a method to measure the performance of the weights. We showed the sample size requirements for the weights and then discussed the curse of dimensionality that occurs as a general problem when balancing multidimensional data. Although the curse of dimensionality remains in our method, we believe that this study provides a basic approach for estimating the balancing weights of multidimensional data using variational $f$-divergence.

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
