# OpenReview forum: "Generalized Balancing Weights via Deep Neural Networks"
_NeurIPS.cc/2023/Conference — Submitted to NeurIPS 2023_

### Official Review · Reviewer_bV7F · 2023-06-29

**Soundness:** 2 fair
**Presentation:** 1 poor
**Contribution:** 3 good
**Rating:** 4
**Confidence:** 2

**Summary:**

This paper proposes a method for estimating the balancing weights in causal effect estimation. The balancing weights are given by the density ratio of the counterfactual distribution and the observed distribution, and the density ratio can be estimated by solving the optimization of the variational expression of the f-divergence between the two distributions. The authors claim that using the $\alpha$-divergence is useful because it addresses the vanishing-gradient problem when we use a neural network for the model. The paper presents some theoretical results.

**Strengths:**

- Expressing $\phi$ as $\exp(T)$ in Eq. (14) is interesting since the domain of the optimization variable becomes simpler.
- The discussion about vanishing gradient is interesting, and the suggested way of setting $\alpha$ may be useful to address this issue.

**Weaknesses:**

- There are several technical parts that might not be precise. See the Questions section below.
- The writing could be improved. The paper rather looks like a collection of definitions of results. Connections between sections are not smooth, and the message of the paper as a whole is not clear.
- The abstract says, "we selected $\alpha$-divergence as it presents efficient optimization because it has an estimator whose sample complexity is independent of its ground truth value and unbiased mini-batch gradients; moreover, it is advantageous for the vanishing-gradient problem." I am not sure what makes the $\alpha$-divergence special in terms of these aspects. Also, the explanation about the vanishing-gradient issue in Section 5.1 is not convincing enough to make me believe that the authors' suggestion really addresses the issue.
- The paper relies on references (including one without open access) for important definitions. The paper could be more self-contained.
- Overall, the paper is mainly about estimating density ratios by the variational expression of the f-divergence, which is not novel.
- There is no experiment in the main part of the paper.

**Questions:**

- Is Definition 3.1 really the same as the standard definition from Pearl (2009)? Which part of the book are the authors referring to?
- I believe there is something wrong with Eq. (4) and Eq. (7). The right hand side depends on $P(\boldsymbol{X}_1), ..., P(\boldsymbol{X}_n)$ while the left hand side does not.
- In lines 207-211, what is $\overset{\to}{\infty}$?
- "We note that the selecting of α does not cause instability in numerical calculations for cases where $E[\nabla_{\theta} \mathcal{L}_{\alpha}(θ)] \to \infty - \infty$." Is there any theoretical result to support this claim?
- Theorem 6.1 looks strange because Eq. (19) concerns $\tilde{P}$ although Eq. (18) has nothing concerning $\tilde{P}$.
- What does it mean by using test data for validation? Is this validation performed only for performance evaluation or also for hyper-parameter tuning?
- As far as I know, people just perform as many epochs as they can afford while keeping the model with the best validation performance. So, fixing the early stopping time is not very useful in practice unless. Am I missing something?
- The $\alpha$-divergence "has an estimator whose sample complexity is independent of its ground truth value": is this true? Eq. (13) seems to depends on both $P$ and $Q$.
- "unbiased mini-batch gradients": isn't this true for any $f$-divergence?

-----
Edit on Aug 2: There is another thing that I would like the authors to address if it is not too late:
Is Eq. (10) correct? Is $f'$ the derivative of $f$ and are we optimizing the objective with respect to $\phi$?

I thought that the authors use the result from [15], but I cannot exactly match the equation with the one in the paper.

[15] XuanLong Nguyen, Martin J Wainwright, and Michael Jordan. Estimating divergence functionals and the likelihood ratio by penalized convex risk minimization. Advances in neural information processing systems, 20, 2007.

**Limitations:**

The authors discuss limitations of the work.

---

> ### Author Rebuttal · Authors · 2023-08-06
>
> We thank the reviewer for a detailed and thoughtful review. We apologize for the delay in responding to your questions.
> We will incorporate other suggestions raised by the reviewer regarding the presentation and other experimental details in the final version. Below we address the main concerns raised in this review.
>
> ***
> ## Questions:
> > Is Definition 3.1 really the same as the standard definition from Pearl (2009)? Which part of the book are the authors referring to?
>
> ### Answer:
> We refered to Equation (3.14) in section 3.2.3.
> Note that  in our definition,  a probability distribution containing all observed and  unobserved variables is considered first,  and the causal effect is defined  by marginalizing  out variables that are not used.
>
> ***
> ## Questions:
> > I believe there is something wrong with Eq. (4) and Eq. (7). The right hand side depends on while the left hand side does not.
>
> ### Answer:
> Please note that Eq. (4) and Eq. (7) are used for different definitions.
>
> ***
> ## Questions:
> > In lines 207-211, what is $\vec{\infty}$?
>
> ### Answer:
> $\vec{\infty}$ denotes $(\infty, \infty, ..., \infty)$.
>
> ***
> ## Questions:
> > Edit on Aug 2: There is another thing that I would like the authors to address if it is not too late: Is Eq. (10) correct? Is  the derivative of  and are we optimizing the objective with respect to ?
> I thought that the authors use the result from [15], but I cannot exactly match the equation with the one in the paper.
>
> ### Answer:
> Eq. (5) in [15]  corresponds to Eq. (10) in the present study.
>
> Note that $\phi$ in [15] corresponds to $f$ in Eq. (10), and
> $\partial \phi$ in [15] to  $f'$ in Eq. (10), here $\partial$ denotes the sub-differential of $\phi$ (not the partial differential of $\phi$),
> and that $q0/p0$ in [15] is identical to $\phi$ in Eq. (10).

---

> > ### Comment · Reviewer_bV7F · 2023-08-15
> > **Re: Rebuttal by Authors**
> >
> > Thank you for the answers. Unfortunately, some of the answers are not clear enough to address the concerns.
> >
> > > We refered to Equation (3.14) in section 3.2.3. Note that in our definition, a probability distribution containing all observed and unobserved variables is considered first, and the causal effect is defined by marginalizing out variables that are not used.
> >
> > The Equation (3.14) in Section 3.2.3 is very different from the definition in Eq. (1) of the paper. Could you write down both equations in the same notation and explain the correspondence precisely?
> >
> > > I believe there is something wrong with Eq. (4) and Eq. (7). The right hand side depends on while the left hand side does not.
> >
> > Let me try to further precise the question: in Eq. (4), the left-hand side represent a probability distribution of $Y$, but the right-hand side represent a probability distribution of $(\boldsymbol{X}, Y)$. So, there must be something wrong. The same goes for Eq. (7).
> >
> > > Eq. (5) in [15] corresponds to Eq. (10) in the present study.
> > >
> > > Note that
> > > in [15] corresponds to in Eq. (10), and in [15] to in Eq. (10), here denotes the sub-differential of (not the partial differential of ), and that in [15] is identical to in Eq. (10).
> >
> > I still see a difference between them: Eq. (10) applies $f'(\cdot)$ to $\phi$, which is not present in Eq. (5) in [15]. Where does this difference come from?

---

> > > ### Author Response · Authors · 2023-08-19
> > >
> > > We thank the reviewer for  the  detailed questions regarding this study.
> > > We apologize for the delay in responding to your comment.
> > >
> > > > The Equation (3.14) in Section 3.2.3 is very different from the definition in Eq. (1) of the paper. Could you write down both equations in the same notation and explain the correspondence precisely?
> > >
> > > Herein, to avoid confusion, we use only the notations used in [15] for the following discussions, except for using the bold font type to write sets of variables and their values. In particular, $D\_{A}$ denotes the domain of $A$, which is the same as $\mathcal{X}_A$ in us.
> > >
> > > First, please note that  (3.14) in [15] defines the causal effects determined by the following $do$ operation:
> > >
> > > $$
> > > \begin{aligned}
> > > &P(x_1,\ldots,x_n|\hat{\mathbf{s}})   \\\\
> > > &= P(X_1=x_1,\ldots,X_n=x_n|do(\mathbf{S}=\mathbf{s})).
> > > \end{aligned}
> > > $$
> > >
> > > Let $\mathbf{PA}\_{\mathbf{S}}= \bigcup\_{X_i \in \mathbf{S}} PA_i$. From Therom 3.2.2 in  [15], we have
> > >
> > > $$
> > > \begin{aligned}
> > > & P(x_1,\ldots,x_n|\hat{\mathbf{s}})\\\\
> > > &= \sum\_{\mathbf{pa}\_{\mathbf{s}} \in D\_{\mathbf{PA}\_{\mathbf{S}}}} P(x_1,\ldots,x_n|\mathbf{s}, \mathbf{pa}\_{\mathbf{s}}) \cdot P(\mathbf{pa}\_{\mathbf{s}})\\\\
> > > &= \sum\_{\mathbf{pa}\_{\mathbf{s}} \in D\_{\mathbf{PA}\_{\mathbf{S}}}}
> > >  \frac{P(x_1,\ldots,x_n, \mathbf{s}, \mathbf{pa}\_{\mathbf{s}})}{P(\mathbf{s} |\mathbf{pa}\_{\mathbf{s}})}.
> > > \end{aligned}
> > > $$
> > >
> > > Here, the last equality is obtained using the following equation:
> > >
> > > $$
> > > \begin{aligned}
> > > &P(x_1,\ldots,x_n|\mathbf{s}, \mathbf{pa}\_{\mathbf{s}}) \cdot P(\mathbf{pa}\_{\mathbf{s}})
> > > =\frac{P(x_1,\ldots,x_n,\mathbf{s}, \mathbf{pa}\_{\mathbf{s}})}{P(\mathbf{s} |\mathbf{pa}\_{\mathbf{s}})}.
> > > \end{aligned}
> > > $$
> > > Now, we note that our definition considers the general case in which other variables from $\mathbf{X}={X_1, X_2, \ldots, X_n}$ exist.
> > > Then, we use the notation $\mathbf{V}$ for all the observed random variables: $\mathbf{X} \subset \mathbf{V}$, and write $\mathbf{V}'$ for $\mathbf{V} \setminus (\mathbf{X} \cup \mathbf{PA}\_{\mathbf{S}} \cup \mathbf{S})$.
> > >
> > > Subsequently,
> > > $$
> > > \begin{aligned}
> > > &P(X_1=x_1,\ldots,X_n=x_n,
> > > \mathbf{V'}=\mathbf{v}'|do(\mathbf{S}=\mathbf{s}))\\\\
> > > &=P(x_1,\ldots,x_n, \mathbf{V'}=\mathbf{v}'|\hat{\mathbf{s}}) \\\\
> > > &=\sum\_{\mathbf{pa}\_{\mathbf{s}} \in D\_{\mathbf{PA}\_{\mathbf{S}}}}
> > >  \frac{P(x_1,\ldots,x_n, \mathbf{V'}=\mathbf{v}'|\mathbf{s}, \mathbf{pa}\_{\mathbf{s}})}{P(\mathbf{s} |\mathbf{pa}\_{\mathbf{s}})}.
> > > \end{aligned}
> > > $$
> > > By marginalizing $\mathbf{V}'$, we obtain
> > > $$
> > > \begin{aligned}
> > > &P(X_1=x_1,\ldots,X_n=x_n|do(\mathbf{S}=\mathbf{s}))\\\\
> > > &=\sum\_{\mathbf{v'} \in D\_{\mathbf{V}}}
> > > P(X_1=x_1,\ldots,X_n=x_n,
> > > \mathbf{V'}=\mathbf{v}'|do(\mathbf{S}=\mathbf{s}))\\\\
> > > &=\sum\_{\mathbf{v'} \in D\_{\mathbf{V}}}  \sum\_{\mathbf{pa}\_{\mathbf{s}} \in D\_{\mathbf{PA}\_{\mathbf{S}}}} \frac{P(x_1,\ldots,x_n, \mathbf{V'}=\mathbf{v}'|\mathbf{s}, \mathbf{pa}\_{\mathbf{s}})}{P(\mathbf{s} |\mathbf{pa}\_{\mathbf{s}})}.
> > > \end{aligned}
> > > $$
> > > From this, we see that (1) in the present study coincides with (3.14) in [15].
> > >
> > > ($\{X_1, X_2, \ldots, X_n\}$ in the above equality corresponds to $\mathbf{Y}$ in (1), and $\mathbf{S}$ to $\mathbf{X}$.)

---

> > > > ### Comment · Reviewer_bV7F · 2023-08-20
> > > > **Definition 3.1**
> > > >
> > > > Thank you for the detailed answer to my question about Definition 3.1. It addressed my concern. I think it is important to mention that the paper adopt Theorem 3.2.2 of Pearl (2009) as the definition.

---

> > > ### Author Response · Authors · 2023-08-19
> > >
> > > We thank the reviewer for the detailed questions regarding this study. We apologize for the delay in responding to your comment.
> > >
> > > > Let me try to further precise the question: in Eq. (4), the left-hand side represent a probability distribution of $\mathbf{Y}$, but the right-hand side represent a probability distribution of  $(\mathbf{X}, Y)$. So, there must be something wrong. The same goes for Eq. (7).
> > >
> > > We provided the definition  of  $\overline{do}$ as a probability distribution of  $(\mathbf{X}, Y)$.
> > > Thus, the left-hand side represents a probability distribution of $(\mathbf{X}, Y)$ not $\mathbf{Y}$.
> > >
> > > Indeed, $P(\mathbf{Y}| do(\mathbf{X}))$ operation defines a (conditional) probability distribution of $\mathbf{Y}$.
> > > However, we believe that this manner has the disadvantages described in line 115-121:  $do(\mathbf{X})$ operation cannot  represent the intervention of the variables that preserve functional relationships.
> > >
> > > To consider operations that preserve the functional relationship, we believe that the definition must define a probability distribution for all variables rather than  a conditional probability on variables of intervention.

---

> > > > ### Comment · Reviewer_bV7F · 2023-08-20
> > > > **Definition of $P(Y | do(X))$**
> > > >
> > > > If the authors intend to define $P(Y | \overline{do}(\boldsymbol{X}))$ as a probability distribution of $(\boldsymbol{X}, Y)$, there seems there is no problem. However, it is hard for a reader to understand that Deinition 3.2 is overriding even the notation $P(Y | \cdot)$ without any explanation. Also, it is very confusing to use the notation $P(Y | \cdot)$ for a joint distribution. I strongly recommend changing the notation to something clearer.

---

> > > ### Author Response · Authors · 2023-08-19
> > >
> > > We thank the reviewer for the detailed questions regarding this study. We apologize for the delay in responding to your comment.
> > >
> > > > I still see a difference between them: Eq. (10) applies  $f'$ to $\phi$, which is not present in Eq. (5) in [15]. Where does this difference come from?
> > >
> > > Eq. (5) in [15] considers a general function class for $f$ that contains non-differentiable functions. Therefore, the sub-differential $\partial f$ was used instead of $f'$.
> > >
> > > As shown in line 175, we assume that $f$ is twice differentiable.
> > > We can then consider $f'$ because  $\partial f=f'$ holds for any differentiable $f$.
> > >
> > > We would like to note that $f$ is assumed to be twice differentiable for  the  strong monotonicity of $f'$, which guarantees the uniqueness of $\phi = dQ/dP$. In addition, the method proposed in [15] considers an optimization of KL divergence that satisfies this assumption.
> > > Therefore, we believe that the assumption is reasonable.

---

> > > > ### Comment · Reviewer_bV7F · 2023-08-20
> > > > **$f'$ in Eq. (10)**
> > > >
> > > > I was not asking about the difference between $\partial f$ and $f'$. I simply did not see anything corresponding to $\partial f$ or $f'$ in the Eq. (5) in [15].
> > > >
> > > > However, I finally managed to understand how the authors got Eq. (10).
> > > > The authors use the fact from Lemma 1: "Equality holds if and only if the subdifferential $\partial \phi(q_0/p_0)$ contains an element of $\mathcal{F}$." I really think the paper should not have omitted this detail.

---

> > > > ### Comment · Reviewer_bV7F · 2023-08-20
> > > > **My current evaluation of the paper**
> > > >
> > > > Overall, the authors follow-up response has addressed my major concerns about the definitions. I would like to raise my score from 3 to 4.
> > > >
> > > > Looking at the supplementary material (although I have not read all of it), the authors may have very interesting theoretical results. However, I think the presentation of the paper (especially in the main manuscript) should be improved a lot before it can be published.
> > > >
> > > > To me, the main contributions seem to be proving some nice properties of the alpha divergence which are useful for general density ratio estimation. I honestly don't know why the authors decided to limit the scope of the paper to the causal effect estimation, which is just one of many applications of density ratio estimation. I think the causal effect estimation part is overly complicating the notation and making the main messages of the paper hard to understand. I would recommend reorganizing the paper in a way that the reader can understand the main contributions before going into the specific application.

---

> > > > > ### Author Response · Authors · 2023-08-20
> > > > >
> > > > > We thank the reviewer for the detailed and constructive feedback.
> > > > > We appreciate the reviewer's time to understand our study.
> > > > >
> > > > > We will incorporate the reviewer's suggestion regarding the scope of the study.

---

> ### Comment · Reviewer_bV7F · 2023-08-21
> **Remaining questions**
>
> If the authors still have time to respond, I encourage them to comment on these points:
>
> > The $\alpha$-divergence "has an estimator whose sample complexity is independent of its ground truth value": is this true? Eq. (13) seems to depends on both $P$ and $Q$
>
> (I should have looked at Eq. (105) instead of Eq. (13))
>
> > "unbiased mini-batch gradients": isn't this true for any $f$-divergence?

---

> > ### Author Response · Authors · 2023-08-22
> >
> > We apologize for not answering the reviewer's question of concern.
> >
> > > The $\alpha$-divergence "has an estimator whose sample complexity is independent of its ground truth value": is this true? Eq. (13) seems to depends on both
> >
> > The term ‘sample complexity’ would have been ambiguous. We use this to mean   ‘the upper bound of the sample sizes to guarantee that the variance of estimators is below a constant’, or, more strictly,  ‘the upper bound of the sample sizes to guarantee that the ratios of the variance of estimators to the truth values are below a constant’.
> >
> > > "unbiased mini-batch gradients": isn't this true for any $f$-divergence?
> >
> > Some previous studies have addressed the problem of biased mini-batch gradients of $f$-divergences ([4], Section 3.2. in [B3]).
> >
> > [4] Bellemare, M. G., Danihelka, I., Dabney, W., Mohamed, S., Lakshminarayanan, B., Hoyer, S., & Munos, R. (2017). The cramer distance as a solution to biased wasserstein gradients. arXiv preprint arXiv:1705.10743.
> >
> > [B3] Belghazi, M. I., Baratin, A., Rajeswar, S., Ozair, S., Bengio, Y., Courville, A., & Hjelm, R. D. (2018). Mine: mutual information neural estimation. arXiv preprint arXiv:1801.04062.

---

> > > ### Comment · Reviewer_bV7F · 2023-08-22
> > > **Unbiasedness**
> > >
> > > Thank you for sharing the references. I think the paper [B3] addresses the biased mini-batch updates for the "Donsker-Varadhan representation" not the $f$-divergence. In fact, they mention in Footnote 6 that
> > > > From the optimization point of view, the $f$-divergence formulation has the advantage of making the use of SGD with unbiased gradients straightforward
> > >
> > > I agree with them because $f$-divergence has the expectation operation in the very outer side of the expression and taking the empirical approximation of this expectation will give us an unbiased estimate no matter what $f$-divergence we use.
> > >
> > > The point I am making here is that the unbiasedness comes as a general property of $f$-divergences, but as a special property of the $\alpha$-divergence. It might make sense for the paper to mention this to justify the choice of $f$-divergence over other divergences, but I don't think the unbiasedness makes the $\alpha$-divergence special compared to other $f$-divergences.

---

### Official Review · Reviewer_LPz4 · 2023-07-02

**Soundness:** 3 good
**Presentation:** 3 good
**Contribution:** 2 fair
**Rating:** 4
**Confidence:** 4

**Summary:**

This paper proposes an approach for estimating balancing weights. The ultimate target is to use these weights for the estimation of the casual effects of interventions. The search of balancing weights are formulated as density ratio estimation. To tackle this problem, the authors employ a variational representation of the $f$-divergence. . The density ratio is modeled by a neural network. Specifically, this paper advocates the use of $\alpha$-divergence, and provide various ways to improve the practical performances of the proposed weights.

**Strengths:**

- This paper successfully makes use of the density ratio learning tools based on $f$ -divergence for causal inference applications.
- This paper summarizes various related existing results on density ratio learning via variational form of $f$ -divergence.
- This paper provides detailed discussion on various practical improvement of the proposed algorithm, which would benefit practitioners.

**Weaknesses:**

- This paper lacks theoretical results on the causal effect estimation (not the density ratio learning), which is regarded as a key target of this work.
	- It is unclear whether the proposed estimator obtain optimal convergence rates and (semi-parametric) efficiency, when degenerated to standard settings like average treatment estimation, and conditional average treatment estimation.
- This work is not impressive in terms of novelty, as many significant components follow more directly from existing work.
	- The variational form of $f$ -divergence and the corresponding estimation of density ratio has been explored by references [15] and [16].
	- The main theoretical result (Theorem 6.1) seems to be a fairly standard dual result (e.g., Theorem 4.4 of https://arxiv.org/pdf/1003.5457.pdf)
- The theoretical results are not stated clearly. Many results are not stated in rigorous statements with careful listing of assumptions. Some seem to be exclude certain important cases like mixture distributions, as claimed to be a major contribution of this proposed work. (see my question below.)

**Questions:**

- When the target causal effect is a scalar, does the proposed weighted estimator achieve $\sqrt{N}$ rate? is the corresponding weighted estimator efficient?
- For conditional treatment estimation, the target quantity is a function, and could be nonparametric in general. Would the weighted estimator converge? Does it converge with optimal rate?
- How to initialize the algorithm?
- How to choose $C$ in (23) in practice?
- Line 277, can the author formalize their claim that $W_1(Q, \hat{Q}_{K_0}^{(N)})$ (i.e., with clear conditions) and provide a proof? Since the estimator is parametrized by neural network, it leads to a non-convex optimization in general. The loss landscape in terms of the neural network parameters seem complicated, and the performance of the estimator resulting from early stopping seems to be related to initialization. (In the supplement (Section C.3), it seems that the authors analyze a different algorithm (instead of Algorithm 1) where the GD is not applied to the neural network parameters.)
- While it is great that the authors provide a detailed discussion of various ways to improve the performance of the proposed estimator in Section 7.1, these strategies seems relatively standard.
- The authors emphasize that their estimator can handle arbitrary mixture of discrete and continuous interventions, but I don't see much discussion/innovation related to this challenge throughout the paper. It seems to me the only trick that is advocated to deal with this challenge is to view the weights as Radon-Nikodym derivatives (which seems to be pretty well known) and then use NN to model. Am I missing anything?
- Also, in the case of mixture interventions, can the authors confirm and discuss the applicability of the theoretical results in Section 7.1 and 8? It seems that the analysis of the early stopping does not apply to the mixture settings (Assumption E4).

**Limitations:**

Section 8 discusses limitations on sample size requirements. The authors claim that the sample size required for controlling the error of $\hat{Q}_{K_0}^{(N)}$ has an exponential dependence on the dimension. Although I have questions regarding this theoretical results (as stated above), I would not be surprised about the curse of dimensionality. However, since the ultimate target is the causal effect, I am more interested in a direct error or sample complexity analysis of the causal effect. The lack of such analysis is a major limitation of this work.

---

> ### Author Rebuttal · Authors · 2023-08-06
>
> We thank the reviewer for a detailed and thoughtful review. We apologize for the delay in responding to your questions.
> We will incorporate other suggestions raised by the reviewer regarding the presentation and other experimental details in the final version. Below we address the main concerns raised in this review.
>
> ## Questions:
> > When the target causal effect is a scalar, does the proposed weighted estimator achieve
>  rate? is the corresponding weighted estimator efficient?
>
> ### Answer
> We apologize if we don't comprehend the reviewer's question correctly.
> We  answer the question below,  understanding that the question concerns a case in which the target causal effect $Y$ is independent of treatments $\mathbf{X}$ and covariates $\mathbf{Z}$.
>
> In this case, $Y$ is independent of the weight $BW(\mathbf{X},\mathbf{Z})$ because the weight is a function of $\mathbf{X}$ and $\mathbf{Z}$.
> Subsequently, $E[Y \cdot BW(\mathbf{X},\mathbf{Z}) | \mathbf{X},\mathbf{Z}] = E[Y]$.
> This implies $\hat{E}[Y \cdot BW(\mathbf{X},\mathbf{Z}) | \mathbf{X},\mathbf{Z}] = \hat{E}[Y]$, where
>  $\hat{E}$ denotes empirical expectation.
> That is, the estimator of the average causal effect using the weight is equal to the sample mean of $Y$, which has $\sqrt{N}$ consistency.
>
> ***
> ## Questions:
> > For conditional treatment estimation, the target quantity is a function, and could be nonparametric in general. Would the weighted estimator converge? Does it converge with optimal rate?
>
> ### Answer
> The reviewer's question is extremely important. We appreciate the suggestion.
>
> We provide the following proposition, which suggests that the  convergence rate of the weighted estimator is equal to the  nonparametric minimax optimal rate for estimating the average causal effects of an intervention.
>
> #### Proposition A1.
> *Let $P$ and $Q$ the probabilities defined in Assumption E4.
>       Let $\hat{Q}^{(N)}\_{K_N}$ denote the probability defined in Proposition C.23.
>       Let $\delta > 0$.
>       Assume for all $N$, there exists $K_N \ge N$, such that $E\_{\mu}\left|\hat{Q}^{(N)}\_{K_N} - Q\right| < \frac{1}{N}^{\frac{1}{d+\delta}}$.
>       Then, it holds for $\delta' > \delta$,
>       $
>       \left| E\_{\hat{Q}^{(N)}\_{K_N}}\left[Y | \mathbf{x} \right]
>               -
>       E\_{Q}\left[Y | \mathbf{x} \right]  \right|
>       = O\left(\frac{1}{N^{\frac{1}{d+\delta'}}} \right)
>       $,  $\mu$-almost everywhere.*
> ***
> * Here, in the satement of Proposition A1, the assumption ❝for all $N$, there exists $K_N \ge N$, such that $E\_{\mu}\left|\hat{Q}^{(N)}\_{K_N} - Q\right| < \frac{1}{N}^{\frac{1}{d+\delta}}$❞ is made not in a probabilistic sence, to simplify the proof described below.
> * The above assumption is based on the following fact, which is obtained using the discussions of the proofs of Proposition C.23, Corollary C.24, and Corollary C.25: Let $\delta > 0$, under Assumpution E1-E6, for a sufficiently large $N$,  there exists $K_N \ge N$ such that
>  $
>         E\_{\mathbf{X}\_P^{(N)}}\left[ E\_{\mu}\left|\hat{Q}^{(N)}\_{K_N} - Q\right| \right]
>          < \frac{1}{N}^{\frac{1}{d+\delta}}
>   $ holds.
>
> From the description in Appendix H of [B1], we see that the weighted estimator achieves the minimax rate for estimating the average causal effects of an intervention when the function space is a subset of the Holder ball with smoothness level $\alpha \le d/(d-2)$.
>
>     [B1] Kallus, N., Mao, X., & Uehara, M. (2021). Causal inference under unmeasured confounding with negative controls: A minimax learning approach. arXiv preprint arXiv:2103.14029.
>
> Finally, we provide the proof of Proposition A1.
>
> ***
> proof of Propositon A1.
> Since $\lim\_{N \rightarrow \infty} E\_{\mu} \left[ N^{\frac{1}{d+\delta'}} \cdot \left|\hat{Q}^{(N)}\_{K_N} - Q\right|  \right] = 0$, we have
>           $N^{\frac{1}{d+\delta'}} \cdot \left|\hat{Q}^{(N)}\_{K_N} - Q\right| \rightarrow 0$ ($\mu$-a.e.), as $N \rightarrow \infty$.
>
> Now, note that
> $$
> \begin{aligned}
>   \left| E\_{\hat{Q}^{(N)}\_{K_N}}\left[Y | \mathbf{x} \right]
>       -
>       E\_{Q}\left[Y | \mathbf{x} \right]  \right|
>   &=
>         \left| E\_{P}\left[Y \cdot \frac{dQ}{dP} \Biggl| \mathbf{x} \right]
>         -
>         E\_{P}\left[Y  \cdot e^{ - T\_{K_N}^{(N)}} \Biggl| \mathbf{x} \right]  \right|
>     \\\\
>   &= \left| E\_{P}\left[Y | \mathbf{x} \right]   \right|
>     \cdot
>     \left|       \frac{dQ}{dP}   - T\_{K_N}^{(N)}   \right| \\\\
> &= \left| E\_{P}\left[Y | \mathbf{x} \right] \right|
>   \cdot
>   \frac{dP}{d\mu} (\mathbf{x})
>     \cdot
>     \left|       \frac{dQ}{dP}   - T\_{K_N}^{(N)}   \right|.
> \end{aligned}
> $$
>
> From this, we have, as $N \rightarrow \infty$,
>       $$
>       \begin{aligned}
>          &N^{\frac{1}{d+\delta'}} \left| E\_{\hat{Q}^{(N)}\_{K_N}}\left[Y | \mathbf{x} \right]
>               -
>               E\_{Q}\left[Y | \mathbf{x} \right]  \right| \\\\
>        &= \left| E\_{P}\left[Y | \mathbf{x} \right] \right|
>         \cdot
>         \frac{dP}{d\mu} (\mathbf{x})
>           \cdot
>           N^{\frac{1}{d+\delta'}}
>           \left|       \frac{dQ}{dP} (\mathbf{x})   - T\_{K_N}^{(N)} (\mathbf{x})  \right|
>           \longrightarrow 0 \ (\mu-\text{a.e.}).
>       \end{aligned}
>       $$
> (Q.E.D)
> ***
> ## Questions:
> > The authors emphasize that their estimator can handle arbitrary mixture of discrete and continuous interventions, but I don't see much discussion/innovation related to this challenge throughout the paper.
>
> ### Answer:
> We developed the Radon-Nikodym derivative estimation method proposed in this study to obtain a method to estimate the average causal effect of simultaneous multidimensional interventions.
>
> We regret that there is less discussion about estimating the average causal effects to explain the method for Radon-Nikodym derivative estimation.
>
> As the reviewer pointed out, we think it would be better for other researchers to split our study into two studies: esitmating the Radon-Nikodym derivatives and estimating the average causal effect of simultaneous multidimensional interventions.

---

> > ### Comment · Reviewer_LPz4 · 2023-08-14
> >
> > Thank you for your reply, and the additional analysis for the nonparametric setting.
> >
> > - To your first response: Thank you for attempting to answer my question. I didn't intend to focus on the specific setting with independence between " $Y$ " and " $X$ and $Z$ ". I was referring to settings such that the estimand (the target causal effect) is a scalar, like average treatment effect (ATE). For instance, many weighted estimators for ATE in the literature are $\sqrt{N}$ -consistent, as well as semi-parametric efficient.
> > - To your second response: Thank you very much for extending your analysis to answer my question. In your statement, you didn't specify $\mathbf{x}$ . The convergence seems to be interpreted on a single **fixed** (but artibrary) $\mathbf{x}$ , and the convergence is pointwise. I would note that common guarantees are usually in $L_2$ , empirical $L_2$ , or uniform sense.
> >
> > Currently, I still think the paper has several issues that would benefit from a more substantial revision and so I intend to keep my original rating. I encourage the authors to keep the momentum and look deeper into the theory of the weighted estimator, which is regarded as the target of the current work, in the future.

---

> > > ### Author Response · Authors · 2023-08-19
> > >
> > > We apologize for the delay in responding to your comment.
> > >
> > > > To your first response: Thank you for attempting to answer my question. I didn't intend to focus on the specific setting with independence between "$Y$" and "$X$" and  "$Z$". I was referring to settings such that the estimand (the target causal effect) is a scalar, like average treatment effect (ATE). For instance, many weighted estimators for ATE in the literature are  $\sqrt{N}$-consistent, as well as semi-parametric efficient.
> > >
> > > Thank you for explaning the settings you refered to.
> > > We consider that even in semi-parametric settings, the weights would only guarantee the same rates as the nonparametric minimax optimal rates of ATE.
> > >
> > > > To your second response: Thank you very much for extending your analysis to answer my question. In your statement, you didn't specify $\mathbf{x}$. The convergence seems to be interpreted on a single fixed (but artibrary) $\mathbf{x}$, and the convergence is pointwise. I would note that common guarantees are usually in $L_2$, empirical $L_2$, or uniform sense.
> > >
> > > Thank you for reviewing the  proposition  in detail and noting the preferable manners to guarantee convergence rates for ATE.
> > > In the case of $L_2$ or uniform sense, the proposition may require additional assumptions. For example, the boundedness of $dQ/dP$ would be sufficient.
> > >
> > > However, we recognize that these discussions should be done carefully and in detail, which would
> > > benefit other researchers.
> > >
> > > We appreciate this detailed review and constructive comments.

---

### Official Review · Reviewer_L7FM · 2023-07-07

**Soundness:** 4 excellent
**Presentation:** 2 fair
**Contribution:** 4 excellent
**Rating:** 4
**Confidence:** 2

**Summary:**

This paper derives a novel method of reweighting covariates for the purpose of causal inference. This is done through learning optimal change of measure weights modeled via neural networks and trained using alpha-divergence measures between the true joint distribution and a mutually independent one. Further tweaks are introduced to help enable this procedure to be more practical in its implementation.

A brief disclaimer: I am not very familiar with causal inference (to be honest I am not quite sure how I received this paper to review); however, I am quite familiar with variational inference and related techniques. I will focus primarily on the latter during my review, but please keep all of this in mind while reading my comments.

**Strengths:**

All of the work presented is very precise and detailed, with novel contributions that I can see with regards to the learning of the weights via the alpha-divergence objective. I found the result of the main algorithm to be particularly concise and intuitive for such an involved derivation (e.g., almost trivial to describe but definitely not to prove). As far as I could tell, all of the decisions made throughout the proposal were well justified and deliberate.

**Weaknesses:**

The precision presented definitely came at the cost of readability in my opinion. The paper is very notation heavy and introduces many concepts rapidly and with great precision that lead me to having to re-read sections many times over to understand the message. It feels like the paper could benefit from summarizing some of the technical details in the main paper and reproduce the more exact version in the appendix.

Additionally, I understand the work faced space limitations; however, I believe it should definitely have the numerical experiments presented in the main paper. If pressing for pages, I could see including the experiments in the main paper and moving section 7.1 (either partially or completely) to the appendix.

I found no weaknesses in the technical information itself from what I could understand.

**Questions:**

The derivations showcased that optimal values of alpha range from 0 to 1, in your experience was there any intuition over what value in this range alpha should be? Or put differently, is there much difference between different values so long as they are all in this range? I am primarily interested both in terms of the resulting divergence value, but also in terms of the resulting downstream effect that such weights would have on applications.

**Limitations:**

The authors very adequately discussed limitations.

---

> ### Author Rebuttal · Authors · 2023-08-06
>
> We thank the reviewer for a detailed and thoughtful review. We apologize for the delay in responding to your questions.
> We will incorporate other suggestions raised by the reviewer regarding the presentation and other experimental details in the final version. Below we address the main concerns raised in this review.
>
> ## Questions:
> > The derivations showcased that optimal values of alpha range from 0 to 1, in your experience was there any intuition over what value in this range alpha should be? Or put differently, is there much difference between different values so long as they are all in this range? I am primarily interested both in terms of the resulting divergence value, but also in terms of the resulting downstream effect that such weights would have on applications.
>
> ### Answer:
> We would like to answer the reviewer's question from a theoretical perspective.
>
> The value of $\alpha$ can be determined by the true divergence value and the number of samples.
>
> Note that the standard deviation of the estimation is smallest when $\alpha=1/2$ (Proposition C.9. in Section C.1), and  that $\alpha$ divergences increase monotonically as the value of $\alpha$ increases (Theorem 3 in [B2]).
>
> Therefore, if the true value of $D\_{1/2} (Q|||P)$ or the number of samples is sufficiently large, then $\alpha=1/2$ is a good choice.
>
> On the other hand, if the true value of  $D\_{1/2} (Q||P)$ is  not  sufficiently large  compared to  the standard deviation of the estimation, the estimated values of the divergence have volatility on the same scale as its value.
> In such cases, by searching $\alpha$ in the range $\alpha > 1/2$, we may find some $\alpha$ such that the estimated values of the divergence are sufficiently larger than its volatility. Specifically, $\alpha$ can be determined as the smallest value such that the following confidence interval does not include zero:
> $$
> \begin{aligned}
>   &\left[ D\_{\alpha} (Q||P) - z\_{1-\beta}\frac{\sigma\_{\alpha}}{\sqrt{N}}, \
>   D\_{\alpha} (Q||P) + z\_{\beta}\frac{\sigma\_{\alpha}}{\sqrt{N}} \right],  \\\\
>  & \text{where}  \  \sigma\_{\alpha} = \sqrt{ C^1\_{\alpha} \cdot D\_{2 \alpha -1}(Q||P)
>         +  C^2\_{\alpha} \cdot  D\_{\alpha}(Q||P) +  C^3\_{\alpha} \cdot  D\_{\alpha}(Q||P)^2 },
> \end{aligned}
> $$
> where $z\_{\beta}$ denotes the $\beta$-th percentile of the normal distribution.
>
> In practice, the true values of $ D\_{2 \alpha -1}(Q||P)$ and $D\_{\alpha}(Q||P)$ are unknown.
> Thus,  after estimating  the divergence  with different $\alpha$,  the optimal $\alpha$ is determined as the closest value to  $\alpha=1/2$, such that the  estimation of  the divergence is sufficiently stable.
>
>     [B2] Van Erven, T., & Harremos, P. (2014). Rényi divergence and Kullback-Leibler divergence. IEEE Transactions on Information Theory, 60(7), 3797-3820.

---

> > ### Comment · Reviewer_L7FM · 2023-08-19
> > **Replay to Authors**
> >
> > My apologies for taking so long to reply to your initial response; I was waiting to read the discussion amongst the other reviewers since I am not the most familiar with causal inference. After having read all of the other reviews and corresponding responses, I defer to what I believe the common thread amongst them which is that there needs to be some major revisions in order to better link the theory developed and the setting of interest, as well as more adequately analyze the performance and success of the method actually being used for causal inference. I do think that the theory developed in this work is promising and should continue to be investigated; however, the current iteration seems to require too major of revisions for this specific submission to be viable.

---

> > > ### Author Response · Authors · 2023-08-19
> > >
> > > We thank the reviewer for the detailed and thoughtful review and appreciate the constructive feedback.
> > >
> > > We will incorporate the reviewer's suggestion regarding the presentation of this study.

---

### Official Review · Reviewer_vQ5H · 2023-07-07

**Soundness:** 2 fair
**Presentation:** 2 fair
**Contribution:** 2 fair
**Rating:** 4
**Confidence:** 1

**Summary:**

This work presents the way to estimate the balance weight, that represents the causal effects of arbitrary mixture of intervention, by using the neural network. Specifically, authors recognize this balance weight as the density ratio between the source and balanced distributions, and estimates this ratio by optimizing the variational representation of  $\alpha$-divergence with $\alpha \in (0,1)$. In this procedure, authors justify why the $\alpha$-divergence should be used in terms of the property of the estimator, such as the sample complexity and unbiasedness, and the vanishing gradient issue for training.



**Strengths:**

* This work seems to have a solid theoretical explanation for the balance weight estimation through neural network.

**Weaknesses:**

First of all, I confess that I am not expert in this field, and thus ask for understanding if my feedback does not make sense in this field.

* The proposed has not been compared with other baselines for balance weights.
> Authors claim that the proposed approach is a general method that can estimate balancing weight even when the datasets are generated from arbitrary mixture of discrete and continuous interventions. This means that the proposed method could be comparable with other baselines if either discrete interventions or continuous interventions exists on datasets. I believe that this comparison seems necessary because it can validate whether the proposed method can be regarded as the general method or not by showing that the proposed method is competitive with the existing methods.  However, since the proposed method was validated only for mixed variable interventions, its effectiveness on single intervention (either discrete or continuous) is not clear.


*  The advantages of the proposed approach seems less clear as compared to the existing baseline.
> In appendix, table 3 shows that the baseline approach (Entropy Balancing) outperforms the proposed method in most cases. Therefore, I am skeptical about why the proposed approach is meaningful. Does the proposed method have any other advantages over the baseline approach, that are not shared at current draft ?

**Questions:**

See above Weaknesses.

**Limitations:**

See above Weaknesses.

---

> ### Author Rebuttal · Authors · 2023-08-06
>
> We thank the reviewer for a detailed and thoughtful review. We apologize for the delay in responding to your questions.
> We will incorporate other suggestions raised by the reviewer regarding the presentation and other experimental details in the final version. Below we address the main concerns raised in this review.
>
> ***
> ## Questions:
> > The proposed has not been compared with other baselines for balance weights.
> > > Authors claim that the proposed approach is a general method that can estimate balancing weight even when the datasets are generated from arbitrary mixture of discrete and continuous interventions. This means that the proposed method could be comparable with other baselines if either discrete interventions or continuous interventions exists on datasets. I believe that this comparison seems necessary because it can validate whether the proposed method can be regarded as the general method or not by showing that the proposed method is competitive with the existing methods. However, since the proposed method was validated only for mixed variable interventions, its effectiveness on single intervention (either discrete or continuous) is not clear.
>
> ### Answer:
> If the purpose of the numerical experiment is to demonstrate the effectiveness of the method,  then the reviewer is correct.
>
> The numerical experiments in Section D.3 were conducted to  observe the curse of dimensionality by confirming the accuracy of the estimation as the data size was changed.
>
> However, to be honest, the experiments were prepared to demonstrate the effectiveness of the method at first, and we acknowledge that it would be inappropriate to observe the curse of dimensionality of the estimation for causal effects.
>
> ***
> ## Questions:
> > > The advantages of the proposed approach seems less clear as compared to the existing baseline.
> > In appendix, table 3 shows that the baseline approach (Entropy Balancing) outperforms the proposed method in most cases. Therefore, I am skeptical about why the proposed approach is meaningful. Does the proposed method have any other advantages over the baseline approach, that are not shared at current draft ?
>
> ### Answer:
> Considering the results of the numerical experiments in Section D.3 and the curse of dimensionality described in our study, the proposed approach has no advantage over the existing methods in terms of estimating causal effects.
>
> However, we believe that the main contribution of this study is to show a posibility of estimating the causal effects of an arbitrary mixture of discrete and continuous interventions.
>
>  Inspired by our study, we expect that someone  appears to develop a method to obtain balancing weights that do not suffer from the curse of dimensionality.

---

### Decision · Program_Chairs · 2023-09-21

**Decision:**

Reject

**Comment:**

The submission utilises a method based on the variational form of f-divergence for density estimation, for estimating balancing weights in causal effect estimation. Most reviewers expressed concerns about the framing of the method, the novelty and the lack of theoretical results for causal effect estimation, and thought the required changes would be substantial to warrant additional reviews. We hope the extensive discussions here will be useful in preparing the next iteration.